# Online Left-Hemispheric In-Phase Frontoparietal Theta tACS for the Treatment of Negative Symptoms of Schizophrenia

**DOI:** 10.3390/jpm11111114

**Published:** 2021-10-29

**Authors:** Chuan-Chia Chang, Cathy Chia-Yu Huang, Yong-An Chung, Jooyeon Jamie Im, Yen-Yue Lin, Chin-Chao Ma, Nian-Sheng Tzeng, Hsin-An Chang

**Affiliations:** 1Department of Psychiatry, Tri-Service General Hospital, National Defense Medical Center, Taipei 11490, Taiwan; changcc@mail.ndmctsgh.edu.tw (C.-C.C.); pierrens@mail.ndmctsgh.edu.tw (N.-S.T.); 2Department of Life Sciences, National Central University, Taoyuan 32001, Taiwan; chuang@ncu.edu.tw (C.C.-Y.H.); yyline.tw@yahoo.com.tw (Y.-Y.L.); 3Department of Nuclear Medicine, College of Medicine, The Catholic University of Korea, Seoul 21431, Korea; yongan@catholic.ac.kr (Y.-A.C.); jooyeonim@gmail.com (J.J.I.); 4Department of Emergency Medicine, Tri-Service General Hospital, National Defense Medical Center, Taipei 11490, Taiwan; 5Department of Emergency Medicine, Taoyuan Armed Forces General Hospital, Taoyuan 32549, Taiwan; 6Department of Psychiatry, Tri-Service General Hospital Beitou Branch, National Defense Medical Center, Taipei 114, Taiwan; touchnature@gmail.com

**Keywords:** transcranial alternating current stimulation, online stimulation, frontoparietal theta coupling, schizophrenia, negative symptoms, working memory

## Abstract

Negative symptoms represent an unmet need for schizophrenia treatment. The effect of theta frequency transcranial alternating current stimulation (theta-tACS) applied during working memory (WM) tasks on negative symptoms has not been demonstrated as of yet. We conducted a randomized, double-blind, sham-controlled trial of 36 stabilized schizophrenia patients, randomized to receive either twice daily, 6 Hz 2 mA, 20 min sessions of in-phase frontoparietal tACS or sham for five consecutive weekdays. Participants were concurrently engaged in WM tasks during stimulation. The primary outcome measure was the change over time in the Positive and Negative Syndrome Scale (PANSS) negative subscale score measured from baseline through to the 1-month follow-up. Secondary outcome measures were other symptom clusters, neurocognitive performance, and relevant outcomes. The intention-to-treat analysis demonstrated greater reductions in PANSS negative subscale scores at the end of stimulation in the active (−13.84%) than the sham (−3.78%) condition, with a large effect size (Cohen’s d = 0.96, *p* = 0.006). The positive effect endured for at least one month. Theta-tACS also showed efficacies for cognitive symptoms, WM capacity, and psychosocial functions. Online theta-tACS offers a novel approach to modulate frontoparietal networks to treat negative symptoms of schizophrenia. The promising results require large-scale replication studies in patients with predominantly negative symptoms.

## 1. Introduction

Negative symptoms (anhedonia, asociality, affective blunting, and amotivation) are strongly correlated with the long-term prognosis of schizophrenia, but effective treatment for negative symptoms is still under investigation [1]. Identifying a treatment target that has a close link to negative symptoms or highly impacts negative symptoms may help to develop an effective therapy to counteract negative symptoms. For example, cognitive impairment has been identified as a potential treatment target, and cognitive remediation (CR) has been proven to improve negative symptoms of schizophrenia [2,3]. 

The working hypotheses link memory problems with negative symptoms [4,5]. Specifically, anhedonia (i.e., the diminished ability to experience pleasure and reduced reactivity to pleasurable stimuli) is one of the core features of behavioral negative symptoms [6]. Problems in working memory (WM) represent a critical neurocognitive deficit of schizophrenia. It has been suggested that WM serves as a potential underlying cognitive mechanism for the recruitment of motivational resources, anticipatory pleasure, and goal-related behaviors [7], and impaired WM may reduce the ability to retrieve and manipulate information to motivate and guide future behaviors, thereby contributing to diminished motivation and pleasure experience [5]. Neuroimaging studies provided robust evidence for the overlap in the activation of brain networks during hedonic processing and WM [8,9]. Furthermore, evidence indicated a correlation between the activity in WM brain networks and the improvement in negative symptoms following antipsychotic treatment, suggesting a mediating effect of WM on negative symptoms improvement in the context of pharmacological intervention [10]. More recently, studies indicated that 20 sessions of WM training (e.g., dual n-back task training), an active component of CR, showed the neural transfer effect to enhance hedonic processing in individuals with high social anhedonia [11,12] and ameliorate hedonic dysfunction in schizophrenia patients with prominent negative symptoms [13].

Research has indicated that CR might best serve in combination with neurophysiological-based interventions that induce neuroplasticity, e.g., non-invasive brain stimulation (NIBS), offering a new approach to treat cognitive deficits in schizophrenia [14,15]. Transcranial alternating current stimulation (tACS) represents a means of NIBS that applies low-intensity sinusoidal electrical currents to the targeted brain regions through the scalp electrodes. Different tACS current intensities (0.5 to 4 mA), stimulation frequencies (0.1 to 80 Hz), electrode montages, phase differences across the stimulation site (if both the target and reference electrodes are in the same phase of the cycle of the current at any given time, the phase difference will be 0 degrees, i.e., in-phase; if the electrodes are in the opposite phase, the phase difference will be 180 degrees, i.e., anti-phase), and with/without DC offsets contribute to different effects on the brain. One main mechanism underlying tACS effects on the targeted brain region is the entrainment of brain rhythms, with resonance happening between the local endogenous oscillations and the applied frequency of stimulation [16]. Since the local brain rhythms vary from resting state to task state, tACS effects on the brain are known to be both frequency-dependent and state-dependent. 

There are a growing number of studies applying tACS during cognitive tasks (often called “online”). Given that the synchronization of frontoparietal regions at theta frequencies dominates during the execution of WM tasks [17,18], online tACS simultaneously applied at a theta frequency over prefrontal and parietal cortices with a 0° phase difference has the potential for facilitating frontoparietal synchronization and eliciting resonance phenomena, which may, in turn, enhance cognitive performance. The cognitive benefits of online theta (6 Hz) in-phase tACS over left frontoparietal regions were validated in healthy populations [17]. An earlier review considered this novel NIBS approach as a potentially powerful treatment for WM deficits in schizophrenia [14]. In a recent case report of schizophrenia, frontoparietal phase coupling artificially induced by multi-session theta frequency tACS in combination with a WM task significantly improved the performance of WM and several other cognitive domains [19]. Recent findings further highlighted that enhancing WM function would transfer to the improved anhedonia in schizophrenia patients with prominent negative symptoms [13]. However, it is unclear whether online in-phase theta-tACS over the left frontoparietal regions (online theta-tACS) to improve WM function could alleviate negative symptoms of schizophrenia. 

The primary aim of our pilot double-blind, randomized controlled trial was to examine the efficacy of online theta-tACS in improving negative symptoms of schizophrenia. As secondary aims, the biomarkers that might be associated with treatment response were also investigated. Furthermore, we examined online theta-tACS tolerability, adverse effects, and efficacies for other clinical outcomes, WM, and other neurocognitive performance. 

## 2. Materials and Methods

### 2.1. Participants

The present study is a double-blind, randomized, sham-controlled trial, approved by the ethics committee of Tri-Service General Hospital, Taipei, Taiwan (ID: 2-106-05-123) and is registered (ClinicalTrials.gov ID: NCT04545294). A schematic overview of the study protocol is shown in Appendix A. The randomization, blinding procedures, allocation concealment, and definition of dropout are described in the Appendix A. The inclusion criteria were: (1) Patients aged 20–65 with DSM-V-defined schizophrenia or schizoaffective disorder; (2) Duration of illness >2 years; (3) Being clinically stable and on an adequate therapeutic dose of antipsychotics for at least 8 weeks before enrolment; (4) Agreement to participate in the study and provide the written informed consent. The exclusion criteria were: (1) Having unstable medical conditions, current psychiatric comorbidity, prominent mood symptoms, or active substance use disorder (in exception to caffeine and/or tobacco); (2) Having a history of seizures, meningitis, or encephalitis; (3) Having contraindications for transcranial electrical and magnetic stimulation; (4) Having a history of intracranial neoplasms or surgery, or a history of severe head injuries or cerebrovascular diseases; (5) Pregnancy or breastfeeding at enrollment; (6) Scalp skin lesions at the area of electrode application. Participants’ antipsychotic medication and dose remained unchanged throughout the trial.

### 2.2. Online Left-Hemispheric In-Phase Frontoparietal Theta (6 Hz) tACS

Theta-tACS was applied during the dual n-back task (see Appendix A), starting at the beginning of each task. In the active theta-tACS condition, sinusoidal tACS was delivered by two battery-operated devices (Eldith DC stimulator Plus, NeuroConn, Ilmenau, Germany) connected with two 4 × 1 wire adaptors (Equalizer Box, NeuroConn), via 10 carbon rubber electrodes (1 cm radius, high-definition 4 × 1 configuration with a gel layer of 2.0 mm), at 6 Hz frequency, 2 mA current intensity without DC offset, with 100 cycles ramp-up/ramp-down and a 0° relative phase, for 20 min, twice-daily on 5 consecutive weekdays. The electrode montages for theta-tACS over left frontoparietal areas were adapted from Polania et al. [17] and are shown in Figure 1. The electrodes of the 1st stimulator were placed at the International 10-10 electrode position F1, F5, AF3, and FC3 (stimulation electrodes) and CPz (return electrode). For the 2nd stimulator, the electrodes were placed at P1, P5, CP3, and PO3 (stimulation electrodes) and FCz (return electrode). The combined impedance of all electrodes was kept below 15 kΩ using a conductive paste, which also held the electrodes in place. A custom-made pulse generator controlled the two stimulators and created an in-phase (synchronous) setup (0° relative phase difference between the output signals of the two tACS-stimulators). The numerical computation of the electric field (Figure 2) was simulated with HD-Explore^®^ (Soterix Medical, New York, NY, USA). Sham stimulation was delivered in the synchronous condition for the initial 30 s of the 2 mA normal-like stimulation. After that, only a tiny current pulse (110 μA over 15 ms) for impedance control took place every 550 ms during the remaining time.

### 2.3. Outcome Assessments 

Detailed definitions of all outcomes are provided in the Appendix A. The primary efficacy outcome was the change over time in the negative symptom subscale score of the Positive and Negative Syndrome Scale (PANSS). Secondary outcomes included the changes over time in PANSS total score, PANSS five-factor subscale score, PANSS two-subdomain score of negative symptoms, Scale for the Assessment of Negative Symptoms (SANS) score, Personal and Social Performance scale (PSP) score, the abbreviated version of the Scale to Assess Unawareness in Mental Disorder in schizophrenia (SUMD) score, and the accuracy of the dual 2-back task (see Appendix A). All the aforementioned assessments were conducted at baseline, the end of stimulation, and 1-week and 1-month follow-ups (Appendix A). Other secondary outcomes were the changes over time in the scores of self-reported measures (Beck Cognitive Insight Scale (BCIS), Self-Appraisal of Illness Questionnaire (SAIQ), the self-reported version of the graphic personal and social performance scale (SRG-PSP), and Schizophrenia Quality of Life Scale Revision Four (SQLS-R4)) and the results of neurocognitive tests, collected at baseline, the end of stimulation, and 1-week follow-up. A well-established tool was used to measure the frequency of adverse effects of online theta-tACS [20]. The payment (USD 17.95) was offered to research participants to reimburse them for the transportation expense of each visit.

### 2.4. Biological Markers

Potential biologic markers were investigated as predictors and mediators of treatment response to online theta-tACS. Two biologic markers have been analyzed so far: heart-rate variability (HRV) and electroencephalography (EEG). The EEG data will be in a separate manuscript. Studies have reported that HRV is significantly correlated both with negative symptoms [21] and with frontoparietal network connectivity [22,23]. Here, we report the methods of collecting and analyzing the data of HRV. In brief, beat-to-beat (RR) interval time series were collected during two different experimental conditions (a 5-min rest period and then a 5-min dual 2-back task period) at baseline, at the end of stimulation, and at 1-week and 1-month follow-ups (Appendix A). Tobacco smoking and caffeinated beverages were prohibited for 12 h before measures. ECG electrodes were placed on bilateral arms just below the elbows, with a ground electrode placed just above the right wrist bone. The lead I electrocardiogram of each patient was taken for 5 min after sitting and having a rest for 20 min in a soundproof, dim-lighted room with thermostatic control (a rest condition). Then, the participant’s electrocardiogram was taken for another 5 min while performing a dual 2-back task (a task condition). The respiration rate of each participant throughout the ECG recording was monitored by using the BioGraph Infiniti system to ensure the respiratory frequency was within the range of 0.15–0.4 Hz. The ECG signals were acquired, stored, pre-processed according to standardized procedures, and processed by an HRV analyzer (LR8Z11, Yangyin Corp., Taipei, Taiwan) [24,25]. The raw data were processed by researchers blinded to the allocation group and not involved in the trial. The computer algorithm identified each QRS complex and rejected any noise or ventricular premature complex according to its likelihood in a standard QRS template. The algorithm also rejected any R-R interval beyond the range of 273–1500 ms to exclude a paroxysmal heartbeat and replaced it with the average of the preceding and following intervals. The ECG signals were sampled with a sampling frequency of 512 Hz and recorded using an 8-bit analog-to-digital converter. The values of stationary R-R intervals were re-sampled and linearly interpolated at a rate of 7.11 Hz to produce continuity in the time domain. Power spectral analysis was performed by using fast Fourier transformation. The direct current component was deleted, and all analyzed signals were truncated into successive 30-s epochs with 50% overlapping. A Hamming window was applied to each time segment for attenuating the leakage effect. The HRV power spectrum was subsequently quantified into the standard frequency-domain measurements, including low-frequency (LF, 0.04–0.15 Hz) and high-frequency (HF, 0.15–0.40 Hz) power. HF power represents a dominant component of HRV and reflects the efferent chronotropic influence of the myelinated vagal pathway, which originates in the nucleus ambiguous, inhibits sympathetic influences on the heart, and leads to rapid, instantaneous changes in heart rate via nicotinic preganglionic receptors on the cardiac sino-atrial (SA) node [26]. The values of HF power were natural logarithms (Ln) transformed before data analyses to obtain a normal distribution (Shapiro–Wilk tested). In the end, two resting-state indices (RR interval_rest_ and HF-HRV_rest_) and two indices of reactivity to dual 2-back tasks (RR interval_task-minus-rest_ and HF-HRV_task-minus-rest_) were derived from the recorded heart rate series. RR interval_task-minus-rest_ can be calculated by mean RR intervals during dual 2-back tasks minus mean RR intervals during resting conditions (ms) and HF-HRV_task-minus-rest_ by HF-HRV during dual 2-back tasks minus HF-HRV during resting conditions (ln(ms^2^)). 

### 2.5. Statistical Analyses

Analyses were conducted with IBM SPSS Statistics 21.0 software (IBM SPSS Inc., Chicago, IL, USA). To compare the between-group differences in characteristics at baseline, Pearson chi-square test or Fisher’s test was used for qualitative variables and Student’s *t*-test or Mann–Whitney test for quantitative variables. In the intention-to-treat (ITT) sample, the missing data were imputed using the last observation non-missing values. Repeated-measures analyses of variance (RMANOVAs) were used to assess the effects of intervention on outcome measures over time, with “time” as the within-group factor (baseline, after the intervention, and follow-up visits) and “treatment group” (active versus sham stimulation) as the between-group factor. The adjustment for any imbalance in the covariates at baseline was done. When significant “time” × “treatment group” interactions were found, the post-hoc Student’s *t*-tests were used to compare the between-group differences. Cohen’s d effect size was calculated for a quantitative measure of the magnitude for the between-group difference, with d = 0.2 (small), 0.5 (medium), and 0.8 (large). Spearman rank correlations were used to analyze the relationships between biomarkers (RR interval and HRV) and treatment response to online theta-tACS. Statistical significance for the results was set at *p* < 0.05 (two-tailed), and the false discovery rate (FDR) method was used to adjust for multiple comparisons.

## 3. Results

### 3.1. Participant Characteristics

Thirty-six patients were randomly allocated to receive active online theta-tACS (n = 18) or sham stimulation (n = 18) (Figure 3: CONSORT Flowchart), and all of them completed 10 sessions of the trial. One participant in the active group dropped out due to withdrawal of consent after completing the 1-week follow-up visit. One participant in the sham group missed the 1-month follow-up visit. The other participants completed the trial without missing visits. The effectiveness of the blinding protocol was satisfactory (see the Appendix A). There were no significant between-group differences in the sociodemographic, clinical characteristics, and performance of neurocognitive tests at baseline, except for PSP global score and SAIQ need for treatment subscale score (Table 1, Appendix A). 

### 3.2. Primary Outcome

The PANSS negative subscale score in the active tACS and sham group was decreased 13.84 ± 9.96 versus 3.78 ± 9.52% shortly after 10-session online stimulation, 13.19 ± 9.49 versus 3.14 ± 9.54% at the one-week follow-up, and 15.04 ± 9.54 versus 4.76 ± 11.27% at the one-month follow-up. There was a significant group-by-time interaction for PANSS negative subscale score (Appendix A); the interaction remained significant after correcting for baseline PANSS negative subscale scores (F_3,31_ = 3.65, *p* = 0.023). Post-hoc analyses showed significant between-group differences at all post-baseline assessments (Figure 4). The negative symptom severity significantly improved at the end of tACS relative to sham treatment (Table 2) with a large effect size (Cohen’s d = 0.96), and the beneficial effect was maintained at the follow-ups. The daily doses of antipsychotic, anticholinergic, antiparkinsonian, and sedative-hypnotic drugs were not associated with the reduction in PANSS negative subscale score at the end of theta-tACS and the follow-ups (Appendix A).

### 3.3. Adverse Events and Safety

The trial showed no major adverse events. There was no significant difference in the frequency of adverse effects between the active and sham groups (Appendix A).

### 3.4. Secondary Outcomes

There were significant group-by-time interactions for PANSS total score, PANSS factor score for negative symptoms (FSNS), PANSS cognitive factor score, PANSS social amotivation score, SANS score, PSP social useful activities, personal and social relationships, global scores, SRG-PSP social useful activities score, self-certainty subscale score of BCIS, and dual 2-back task accuracy (Appendix A). Post-hoc analyses showed greater improvements at the end of stimulation in active theta-tACS versus the sham group for the abovementioned outcomes, except for PSP personal and social relationships (Table 2 and Table 3). We observed no significant group-by-time interactions for other secondary outcomes. In each of the two trial groups, the degree of decrease in PANSS negative subscale scores was not associated with the improvement in dual 2-back task accuracy.

### 3.5. Biologic Marker

Appendix A showed the results of resting indices (RR interval_rest_ and HF-HRV_rest_) and reactivity indices (RR interval_task-minus-rest_ and HF-HRV_task-minus-rest_) over time for the active versus sham groups. All the indices at baseline failed to show any significant between-group difference (Appendix A). Changes from baseline in all these indices at each postbaseline assessment between the active and sham groups were not significantly different (Appendix A). Correlation analyses in the active group showed that more negative values of baseline RR interval_task-minus-rest_ (i.e., heightened heart rate reactivity to a dual 2-back task) predicted greater reductions in PANSS negative subscale scores at the end of stimulation (r = 0.76, *p* < 0.001; Appendix A and Figure 5A). The increases in RR interval_task-minus-rest_ from baseline to the end of stimulation (i.e., reductions in heart rate reactivity to a dual 2-back task over time) were correlated with higher degrees of decrease in PANSS negative subscale scores during the same period (r = −0.79, *p* < 0.001; Appendix A and Figure 5B). These correlations did not significantly change when outliers were dropped. The statistical significance of other correlation analyses did not survive FDR correction for multiple tests (Appendix A).

## 4. Discussion

A few case reports indicated 1–5 sessions of online in-phase theta-tACS over left frontoparietal regions improved the performance of WM and several other cognitive domains in schizophrenia patients [19,27]. This study is the first randomized controlled trial supporting the efficacy of online theta-tACS in improving negative symptoms of stabilized patients with schizophrenia. The efficacy could be maintained at the 1-month follow-up. Furthermore, the intervention was safe and well-tolerated.

Our study showed online theta-tACS was superior to sham in improving negative symptoms, cognitive symptoms, as well as WM capacity. Furthermore, greater improvements in these deficits translated to more gains in psychosocial functions for the patients. One potential neuronal mechanism for theta-tACS to improve negative symptoms is the entrainment of brain network oscillations [16]. Negative symptoms are associated with cognitive deficits and dysregulated dopaminergic transmission in the mesocorticolimbic pathway (ventral tegmental area (VTA), ventral striatum (VS), hippocampus (HP), and prefrontal cortex (PFC)). An erroneous functional coupling between the PFC, VTA, and HP plays a critical role in these abnormalities [28,29]. Research indicates that theta-rhythm oscillations coordinate neuronal activity in the PFC-VTA-HP axis during information processing (e.g., WM) [30,31]. Moreover, further evidence suggests a potential link between theta oscillations, negative symptoms, and dopamine release in PFC, HP, and subcortical regions. Specifically, the application of rTMS (repetitive transcranial magnetic stimulation) at a theta rhythm (intermittent theta-burst stimulation, iTBS) over the left DLPFC can mitigate negative symptoms of schizophrenia, as well as modulate the neural transmission of PFC, VS, and HP [32,33]. Twenty-session, 20-min offline (without performing a cognitive task during the stimulation) theta-tACS over PFC has been reported to improve negative symptoms of schizophrenia in a case series [34]. Taken together, the neuronal mechanisms for the clinical efficacy of the application of tACS tuned at theta frequency to schizophrenia patients during a resting state may involve entrainment of endogenous oscillation in the brain networks to the stimulation frequency and coupling of long-range oscillatory connectivity between distant brain regions (e.g., the PFC-VTA-HP axis).

The online paradigm of theta-tACS applied in this study may have an additional neuronal mechanism that underlies its clinical efficacy: rhythm resonance, which occurs when the tACS frequency is equal to that of the endogenous brain oscillations [16]. Research suggests that “offline” stimulation depends on the modification of neural activity that persists beyond the duration of stimulation, while “online” stimulation takes effect by modulating a specific network involved in the task [35]. The current study found that online theta-tACS applied during dual n-back training improved WM capacity and accuracy to a larger extent than sham stimulation during that training (Figure 3 and Table 2). The dual n-back task required the participants to simultaneously maintain the auditory and visual n-back streams, and independently monitor and update these stimuli, and continuously adjust to the increasing WM load. Evidence showed that schizophrenia patients had reduced activation in frontoparietal regions during the dual n-back task [36] and that dual n-back training could enhance WM performance and neural efficiency of the frontoparietal network that underlies WM functioning in both healthy individuals and schizophrenia patients [13,37,38]. Impaired WM plays a critical role in the pathophysiology of the social amotivation dimension of negative symptoms, e.g., diminished social drive and anticipatory pleasure [5,7,39]. Research indicated that WM training (e.g., dual n-back training) potentially counteracted negative symptoms, possibly through exerting neuroplastic effects on hedonic processing and normalizing the aberrant brain activation during that process [13]. Specifically, dual n-back training improved WM capacity, hedonic deficits, and inattention symptoms in schizophrenia patients with predominantly negative symptoms. After that training, the frontal brain and insula cortex showed increased activation during the active processing of positive affective incentives [13]. The increased activation in the two brain regions thought to be involved in the motivation salience/value system [40] was correlated with reductions in negative symptoms. Taken together, online theta-tACS may resonate with theta phase oscillatory synchronization across the frontoparietal network elicited by the execution of WM tasks [17], thereby potentiating the efficacy of dual n-back training in improving hedonic process and behavioral negative symptoms (as indexed by the reduction in social amotivation score). Additionally, research has demonstrated that the prefrontal cortex and posterior parietal cortex are mutually and extensively connected with the thalamus, the pulvinar nucleus in particular [41]. In normal brains, the pulvinar serves as a subcortical hub for the functioning of the frontoparietal network and plays an important role in the contextual and multi-sensory processing [42,43,44], and also an emotional response, given its projection to the amygdala [45]. Given its strategic position in sensory and emotional processing, pulvinar has been proven to be involved in the psychopathological symptoms of schizophrenia, e.g., impairment of sensory processing and spatial working memory [46,47]. It is possible that online theta-tACS stimulating the key nodes in the frontoparietal network may normalize the functional connectivity between the cortical and subcortical hub (i.e., pulvinar) regions and thereby contribute to its clinical efficacy. 

Our trial showed that the efficacy of online frontoparietal theta-tACS endured for at least one month beyond the intervention. The prolonged after-effects of tACS may be related to a phenomenon called spike-timing-dependent plasticity (STDP), a process that modifies the connection strengths based on the relative timing of the input and output spikes (or action potentials) of a neuron [16]. STDP means that the synapse will be strengthened if input action potentials occur immediately before the output action potentials, and the synapse will be made weaker if input action potentials occur immediately after output action potentials. The STDP process is known to explain, in part, long-term depression (LTD) and long-term potentiation (LTP) of nervous systems. To sum up, tACS, at a frequency close to that of resonance frequency during stimulation, may intensify the synapses across the stimulated regions through the mechanism of STDP [48,49]. Repetitive (multiple-session) tACS during specific time intervals may consolidate the neuroplasticity effects and may, in turn, elicit a long-lasting effect for further clinical application in treating neuropsychiatric disorders, such as schizophrenia [16,50].

Our results provide evidence that the RR interval_task-minus-rest_ (heart rate reactivity to WM task) could serve as a predictive biomarker for treatment response to online theta-tACS (Figure 5A). Neuroimaging research indicates that activity in the frontoparietal network is engaged by a high WM load [51]. A recent study reported that schizophrenia patients performed worse on a dual 2-back task as well as showed more dysfunctional network activity in frontoparietal cortices compared to healthy subjects [36], suggesting the importance of functional integrity of frontoparietal network for individuals to demonstrate efficient encoding and maintaining WM contents upon performing a high-load WM task. It is known that individuals who perceive the difficulty of a high-load WM task may react to these sustained heavy-task demands by a psychophysiological defense reaction, which is supposed to be derived from the deactivation of the vagal system and activation of the sympathetic nervous system (i.e., cardiovascular arousal: shortening of RR interval (an increase in heart rate) and a reduction in HRV) [52]. Research also indicates that regional cerebral blood flow in frontoparietal cortices positively correlates with the activity of the cardiac parasympathetic (cardiovagal) nervous system, which serves as a brake to suppress the intrinsic sinoatrial node rate and dampen cardio-acceleratory circuitry [51]. It is plausible that patients showing baseline heightened heart rate reactivity to dual 2-back tasks may represent a subpopulation with greater dysfunction within the frontoparietal network while online frontoparietal theta-tACS may be especially effective as an adjunct treatment to remediate the underlying neural impairments of this subset of patients, thereby reducing behavioral and overall negative symptoms. Furthermore, lessened heart rate reactivity to WM tasks over time was associated with negative symptom improvement, which is in line with our putative model proposing that heightened heart rate reactivity to a WM task occurs secondarily to frontoparietal network dysfunction and, therefore, the negative symptom improvement is accompanied by the normalization of heart rate hyper-reactivity in response to a WM task, thus supporting the value of RR interval_task-minus-rest_ as a surrogate biomarker for the clinical efficacy of online theta-tACS (Figure 5B). Taken together, our results provide a novel model for developing a personalized online theta-tACS to treat negative symptoms by applying peripheral biomarkers, e.g., cardiac interbeat interval (RR) time series. 

Our study had limitations. First, our trial did not explicitly target negative symptoms due to the inclusion of a small sample of patients not fulfilling the criteria of predominantly negative symptoms [40]. Further studies should be carried out to examine whether our positive results can be replicated in a larger homogenous sample of schizophrenia patients with predominantly negative symptoms [53]. Second, the PANSS emotional/depressed subscale was used to assess depressive symptom severity over time. It would have been ideal to use the Calgary depression rating scale to better control for the improvement in mood symptoms. Third, there was no evidence for the generalization of the improvement in WM capacity driven by online frontoparietal theta-tACS during dual n-back training onto other untrained cognitive domains. Longer-term follow-up assessments may be required to determine whether there is a delayed onset of gains for other neurocognitive performance [54]. Finally, the WM training protocol in our trial (5-day, twice-daily, dual n-back training for 10 sessions in total) was adapted from Jaeggi et al. [37], who adopted 20-day, once-daily dual n-back training for a total of 20 sessions. Given a positive correlation between a greater number of cognitive training sessions and better performance/positive transfer effects [55], it deserves further investigation into whether more sessions of WM training in combination with online theta-tACS drive generalizing gains in neurocognition, as well as greater improvement in negative symptoms. 

## 5. Conclusions

Our pilot study shows that online frontoparietal theta-tACS during WM training can improve negative symptoms and other clinical outcomes, possibly through modulating the left frontoparietal network. The novel intervention opens a new area of tACS research for negative symptoms in schizophrenia. The precise mechanism of action also needs to be elucidated by future functional neuroimaging studies.

## Figures and Tables

**Figure 1 jpm-11-01114-f001:**
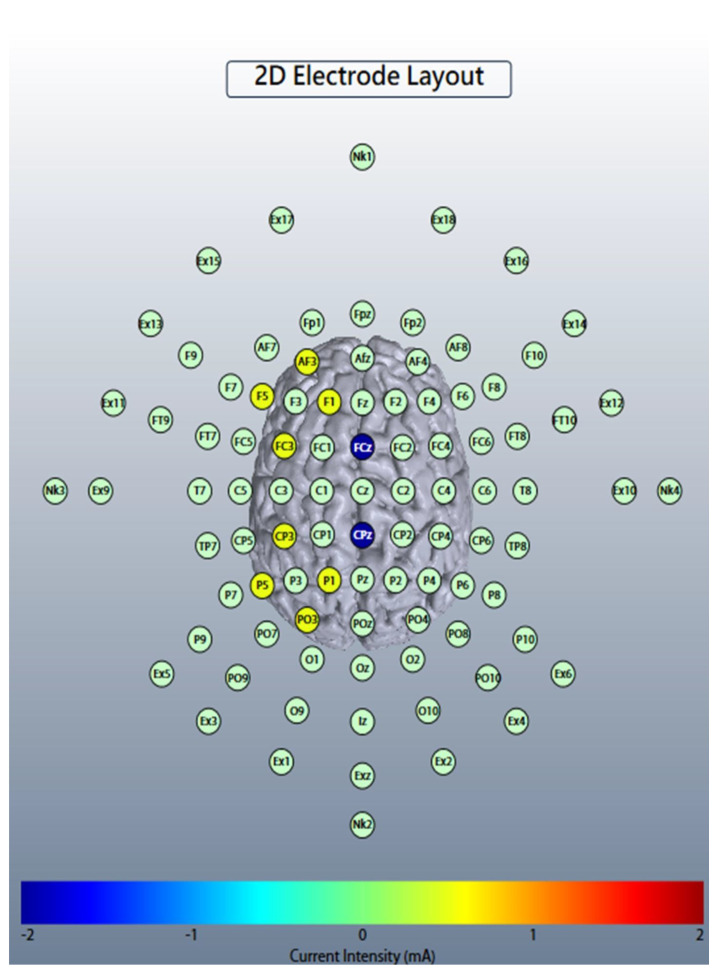
The 2D head model of left-hemispheric in-phase theta-rhythm frontoparietal high-definition transcranial alternating current stimulation (tACS). The electrodes of the first DC stimulator were placed at the International 10-10 electrode position F1, F5, AF3, and FC3 (stimulation electrodes) and CPz (return electrode). For the second stimulator, the electrodes were placed at P1, P5, CP3, and PO3 (stimulation electrodes) and FCz (return electrode). A custom-made pulse generator controlled the two stimulators and created an in-phase (synchronous) setup (0° relative phase difference between the output signals of the two tACS-stimulators). The color bar indicates the estimated current intensity that the electrodes of the stimulators receive.

**Figure 2 jpm-11-01114-f002:**
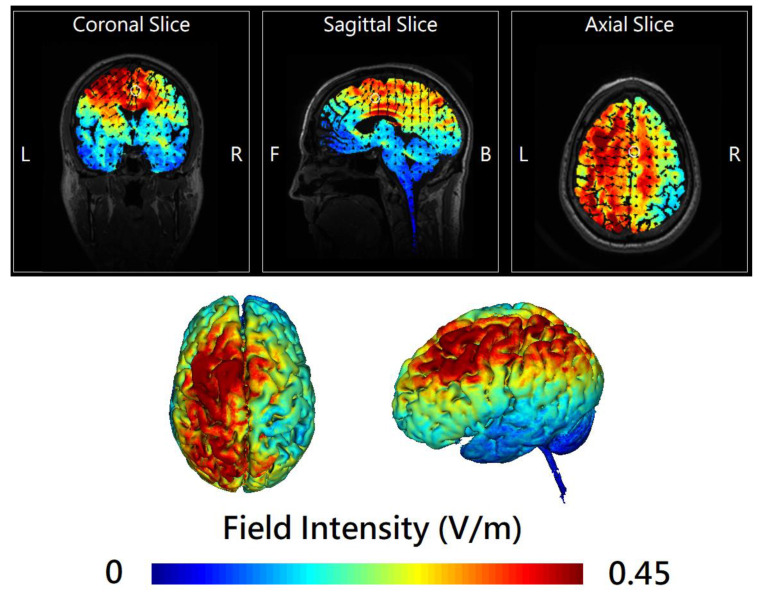
The 2D (upper panel) and 3D (lower panel) representation of electric field simulation of left-hemispheric frontoparietal theta in-phase transcranial alternating current stimulation (tACS) by HD-Explore^®^ (Soterix Medical, New York, NY, USA), which utilizes a finite element model of brain current flow based on an MRI-derived template head. Black arrows in the upper panel represent the vectors of electrical current flow. The color bar indicates the intensity of the electrical field (V/m).

**Figure 3 jpm-11-01114-f003:**
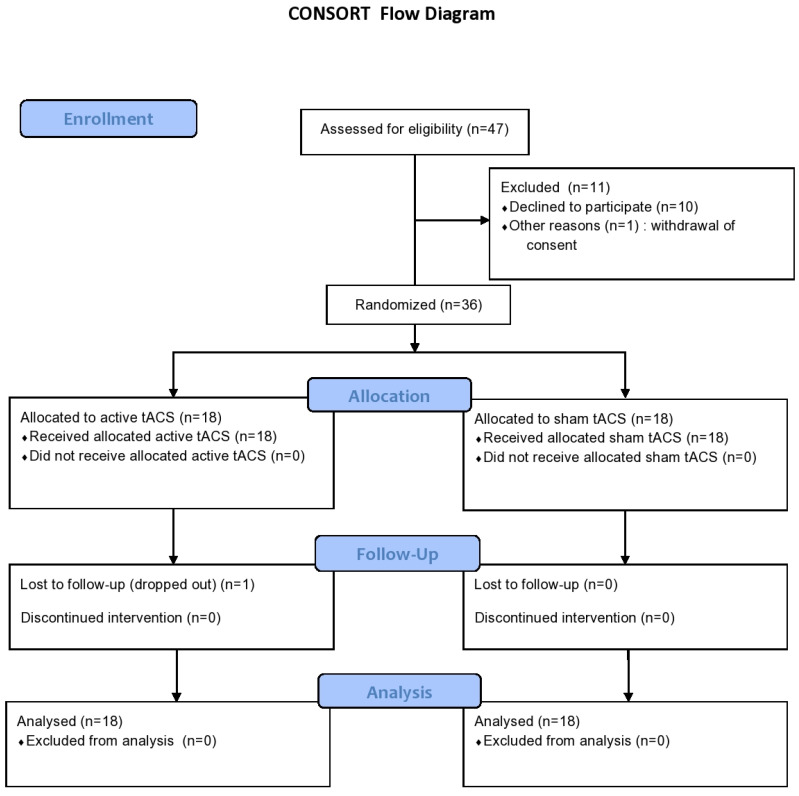
The CONSORT flow diagram of the progress through the phases of the randomized sham-controlled trial.

**Figure 4 jpm-11-01114-f004:**
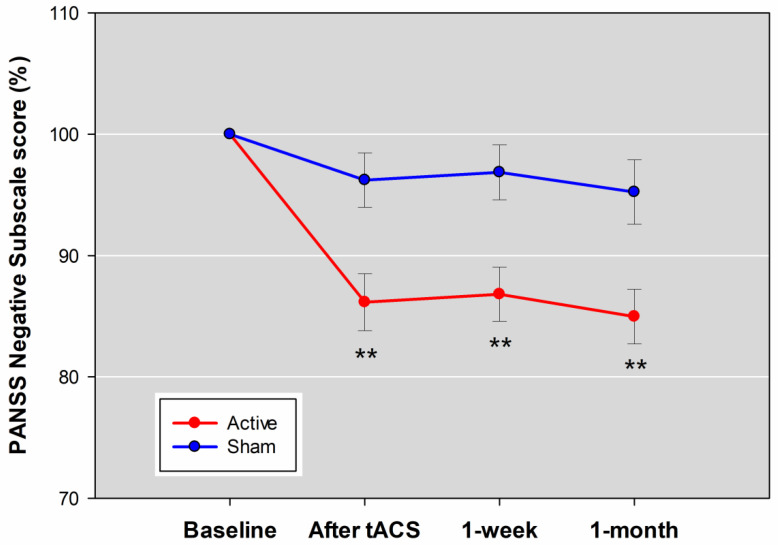
Score as a percentage of baseline in the Positive and Negative Syndrome Scale (PANSS) negative subscale score across the four visits. Post-hoc analyses were conducted to test the differences between the active and sham groups at each post-baseline visit with a *p*-value < 0.05 considered significant. Error bars indicated standard errors. ** *p* < 0.01.

**Figure 5 jpm-11-01114-f005:**
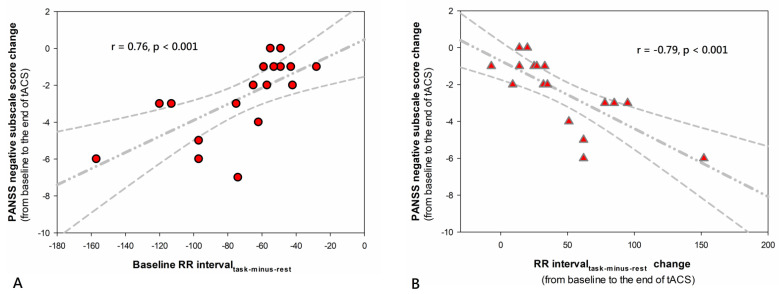
Correlation analyses in patients treated with active online theta-tACS showed that (**A**) more negative values of baseline RR interval_task-minus-rest_ were associated with greater reductions in PANSS negative subscale score from baseline to the end of stimulation, and (**B**) increases in RR interval_task-minus-rest_ from baseline to the end of stimulation were associated with greater decreases in PANSS negative subscale scores during the same period. RR interval_task-minus-rest_ represents RR intervals during dual 2-back tasks minus RR intervals during resting conditions (ms). The regression line and 95% confidence intervals for the linear regression slope are shown.

**Table 1 jpm-11-01114-t001:** Baseline sociodemographic and clinical characteristics of the participants.

Characteristics	Active tACS	Sham	t/U or χ^2^/Fisher’s	*p*-Value
	(n = 18)	(n = 18)		
Schizophrenia/schizoaffective disorder	13/5	15/3	0.64	0.69
Females, n (%)	8 (44.40)	10 (55.60)	0.44	0.51
Age, years	41.78 ± 8.84	43.17 ± 11.20	0.41	0.68
Education level, years	14.44 ± 3.13	12.67 ± 2.79	112.5	0.1
BMI, kg/m^2^	26.50 ± 6.14	27.30 ± 5.37	141	0.51
Weekly regular exercise, hours	2.63 ± 3.79	2.45 ± 2.67	154	0.8
Handedness (right-left)	17/1	15/3	1.13	0.6
Smoker, n (%)	8 (44.40)	3 (16.70)	3.27	0.15
Hypertension, n (%)	3 (16.70%)	3 (16.70%)	0	1
Diabetes mellitus, n (%)	5 (27.80%)	1 (5.60%)	3.2	0.18
Onset age, years	27.44 ± 6.99	25.94 ± 7.44	−0.62	0.54
Length of illness, years	15.28 ± 10.31	17.28 ± 10.58	136.5	0.42
Antipsychotic dosage, mg/day ^a^	21.50 ± 14.04	19.03 ± 13.46	177	0.65
Anticholinergic dosage, mg/day ^b^	0.72 ± 1.60	1.06 ± 1.47	136.5	0.42
Sedative-hypnotic dosage, mg/day ^c^	37.22 ± 42.50	13.89 ± 17.37	200	0.24
PANSS total score	72.44 ± 9.72	74.11 ± 7.30	0.58	0.57
PANSS negative subscale score	19.22 ± 3.86	19.83 ± 3.63	0.49	0.63
PANSS five-factor score				
PANSS-FSNS	21.67 ± 4.96	21.94 ± 4.18	0.18	0.86
PANSS-FSPS	13.56 ± 5.22	13.94 ± 3.93	148.5	0.67
Excited	5.78 ± 2.34	5.83 ± 1.98	147	0.62
Cognitive	9.94 ± 1.70	10.50 ± 1.69	−0.98	0.33
Emotional/depressed	6.78 ± 2.46	6.39 ± 1.58	161.5	0.99
PANSS two-subdomain score of negative symptoms				
Exp Neg	16.50 ± 3.59	17.17 ± 3.20	145.5	0.6
Soc Amot	8.94 ± 2.39	9.11 ± 1.60	0.25	0.81
SANS	51.94 ± 11.87	52.61 ± 10.05	−0.18	0.86

Abbreviations: tACS, Online theta transcranial alternating current stimulation; BMI, Body Mass Index; PANSS, Positive and Negative Syndrome Scale; FSNS, Factor Score for Negative Symptoms; FSPS, Factor Score for Positive Symptoms; Exp Neg, Expressive Negative symptoms; Soc Amot, Social Amotivation; SANS, Scale for the Assessment of Negative Symptoms. Notes: Data are presented as means ± standard deviations unless otherwise stated; Significant *p*-values are presented in bold. ^a^ The daily dose of antipsychotic medications was converted to olanzapine equivalent. ^b^ The daily dose of anticholinergic antiparkinsonian medications was converted to biperiden equivalent. ^c^ The daily dose of sedative-hypnotics was converted to diazepam equivalent.

**Table 2 jpm-11-01114-t002:** Changes in the primary and secondary outcomes after 10 sessions of tACS (n = 18) or sham stimulation (n = 18) in the participants.

	Active tACS	Sham	Cohen’s d	95% CI	*p* ^a^
Variables	Mean ± SD	Mean ± SD			
**Primary outcome**					
PANSS negative subscale score	−2.67 ± 2.14	−0.72 ± 1.84	0.96	0.27 to 1.65	**0.006**
**Secondary outcomes**					
PANSS total score	−5.28 ± 3.98	−0.89 ± 4.39	1.02	0.33 to 1.72	**0.003**
PANSS five-factor score					
PANSS-FSPS	−0.22 ± 0.73	0.06 ± 0.64	0.4	−0.26 to 1.06	0.23
PANSS-FSNS	−2.44 ± 2.59	−0.56 ± 1.85	0.82	0.14 to 1.50	**0.017**
Excited	−0.11± 0.32	0.11± 0.47	0.54	−0.12 to 1.20	0.11
Cognitive	−1.83 ± 1.15	−0.56 ± 1.29	1.02	0.32 to 1.71	**<0.001**
Emotional/depressed	−0.11 ± 0.58	0.28 ± 1.13	0.42	−0.24 to 1.09	0.2
PANSS two-subdomain score of negative symptoms					
Exp Neg	−0.78 ± 1.00	−0.56 ± 1.15	0.2	−0.46 to 1.50	0.54
Soc Amot	−2.17 ± 2.62	−0.39 ± 1.42	0.82	0.15 to 1.51	**0.016**
SANS score	−7.11 ± 5.65	−2.17 ± 4.95	0.91	0.22 to 1.60	0.008
Dual 2-back task accuracy, %	30.39 ± 14.70	10.50 ± 10.00	1.55	0.80 to 2.29	**<0.001**

Abbreviations: tACS, Online theta transcranial alternating current stimulation; PANSS, Positive and Negative Syndrome Scale; FSNS, Factor Score for Negative Symptoms; FSPS, Factor Score for Positive Symptoms; Exp Neg, Expressive Negative symptoms; Soc Amot, Social Amotivation; SANS, Scale for the Assessment of Negative Symptoms. ^a^ Two-tailed Student’s *t*-test. *p*-values are in bold if the primary outcome (PANSS negative subscale score) reaches the significance level of <0.05 or the secondary outcomes reach the corrected significance level (false discovery rate method).

**Table 3 jpm-11-01114-t003:** Changes in other secondary outcomes of participants treated with 10 sessions of active tACS (n = 18) or sham stimulation (n = 18).

Variables	Active tACS	Sham	Cohen’s d	95% CI	*p* ^a^
	Mean ± SD	Mean ± SD			
**Other secondary outcomes**					
PSP					
Social useful activities	−0.44 ± 0.51	−0.11 ± 0.32	0.76	0.08 to 1.43	**0.03**
Personal and social relationships	−0.17 ± 0.38	0.06 ± 0.42	0.56	−0.10 to 1.23	0.09
Self-care	−0.28 ± 0.46	0.00 ± 0.69	0.47	−0.20 to 1.13	0.16
Disturbing and aggressive behavior	−0.28 ± 0.46	−0.17 ± 0.51	0.22	−0.43 to 0.88	0.5
Global score	3.72 ± 3.49	0.56 ± 2.83	0.97	0.28 to 1.66	**0.01**
SUMD					
Awareness of disease	−0.62 ± 2.62	0.00 ± 0.00	0.33	−0.33 to 0.98	0.32
Awareness of positive symptoms	−0.00 ± 3.81	0.00 ± 0.00	0	−0.65 to 0.65	1
Awareness of negative symptoms	−3.09 ± 9.18	−0.62 ± 2.62	0.36	−0.30 to 1.02	0.28
SRG-PSP					
Social useful activities	0.78 ± 1.73	−1.11 ± 2.78	0.8	0.12 to 1.48	**0.02**
Personal and social relationships	−0.28 ± 2.14	0.61 ± 2.30	0.39	−0.27 to 1.05	0.24
Self-care	−0.11 ± 1.64	−0.06 ± 2.07	0.03	−0.63 to 0.68	0.94
Disturbing and aggressive behavior	−0.22 ± 0.94	−0.17 ± 2.43	0.03	−0.63 to 0.68	0.94
Global score	0.61 ± 4.13	−0.39 ± 6.40	0.18	−0.47 to 0.84	0.58
SAIQ					
Total score	0.17 ± 4.62	−3.83 ± 9.13	0.54	−0.12 to 1.21	0.11
Worry	−0.11 ± 3.92	−2.28 ± 5.75	0.43	−0.23 to 1.09	0.19
Need treatment	0.33 ± 1.71	−1.00 ± 2.17	0.67	−0.01 to 1.34	0.05
Presence/outcome	−0.06 ± 2.51	−0.67 ± 3.68	0.19	−0.47 to 0.84	0.57
BCIS					
BCIS-R	−1.17 ± 3.40	1.72 ± 5.53	0.62	−0.05 to 1.28	0.07
BCIS-C	−1.00 ± 2.57	1.78 ± 2.78	1.02	0.32 to 1.71	**<0.001**
R-C index	−0.17 ± 3.73	−0.06 ± 6.30	0.02	−0.63 to 0.67	0.95
SQLS-R4					
Total	−6.83 ± 13.60	−2.61 ± 18.09	0.26	−0.40 to 0.91	0.43
Psychosocial	−4.39 ± 9.62	−0.78 ± 11.07	0.34	−0.32 to 1.00	0.3
Vitality	−2.44 ± 6.31	−1.83 ± 7.84	0.08	−0.57 to 0.74	0.8

Abbreviations: tACS, Online theta transcranial alternating current stimulation; PSP, Personal and Social Performance scale; SRG-PSP, The self-reported version of the graphic Personal and Social Performance scale; SUMD, The abbreviated version of the Scale to Assess Unawareness in Mental Disorder; SAIQ, The Taiwanese version of Self-Appraisal of Illness Questionnaire; BCIS, The Taiwanese version of Beck’s Cognitive Insight Scale; BCIS-R, Self-reflectiveness subscale of BCIS; BCIS-C, Self-certainty subscale of BCIS; SQLS-R4, the Schizophrenia Quality of Life Scale Revision Four. *p*-values are in bold if the corrected significance levels are met (false discovery rate method).

## Data Availability

The data presented in this study are available on request from the corresponding author.

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
