# Peer review of "Online Left-Hemispheric In-Phase Frontoparietal Theta tACS for the Treatment of Negative Symptoms of Schizophrenia"

_jpm, 2021, doi:10.3390/jpm11111114_

Round 1
Reviewer 1 Report
This is an interesting manuscript describing a thoroughly conducted trial investigating a novel hypothesis and treatment methodology in patients with difficult to treat symptoms. There are a number of things that could improve the manuscript:
- Given the proposed justification for trial was that changes in working memory would be modulated with this form of stimulation and that would improve negative symptoms, I would like to have seen some sort of investigation of the relationship between improvement in working memory and improvement in negative symptoms.
- It would have been ideal to use the Calgary depression rating scale to control for improvement in mood
- Why was the success of blinding not reassessed at the end of treatment?
- Was the presence of negative symptoms and their severity part of the inclusion criteria?
- Given th inclusion of patients with schizoaffective disorder, it would commenting on whether patients were excluded who were likely to have significant depressive symptoms (these are not really well assessed on the PANSS emotional/depressed subscale)
- How much payment was provided?
- The use if the HRV biomarker requires some justification
- A mixed model approach would have been better repeated measures one but given the dropout rates I think this is oka
Author Response
Reviewer 1
Comments and Suggestions for Authors
This is an interesting manuscript describing a thoroughly conducted trial investigating a novel hypothesis and treatment methodology in patients with difficult to treat symptoms. There are a number of things that could improve the manuscript:
Overall response to the Reviewer 1
We would like to thank the reviewer for the insightful comments. These comments would be responded one by one as follows.
Comment:1.
Given the proposed justification for trial was that changes in working memory would be modulated with this form of stimulation and that would improve negative symptoms, I would like to have seen some sort of investigation of the relationship between improvement in working memory and improvement in negative symptoms.
Response:1.
As suggested, we revise it on page 11 of the manuscript.
In each of the two trial groups, the degree of decrease in PANSS negative subscale scores were not associated with the improvement in dual 2-back task accuracy.
Comment:2.
It would have been ideal to use the Calgary depression rating scale to control for improvement in mood
Response:2.
As suggested, we revise it on page 15 of the manuscript.
Second, the PANSS emotional/depressed subscale was used to assess depressive symptom severity over time. It would have been ideal to use the Calgary depression rating scale to better control for the improvement in mood symptoms.
Comment:3.
Why was the success of blinding not reassessed at the end of treatment?
Response:3.
We did not report that in the manuscript because the results were similar and seemed redundant. As suggested, we provide the information of guessing active for the two trial groups at the end of treatment on page 2 in the Supplementary Materials.
Effectiveness of blinding
Each participant was asked to answer the question of whether he or she had received active tACS or sham stimulation shortly after the 1st session of stimulation and at the end of treatment. After the trial was unblinded, analyses showed that 83.3% of participants receiving active tACS and 72.2% of those receiving sham guessed they had received active stimulation shortly after the 1st session of stimulation. Fisher's exact test revealed no statistically significant between-group differences (p = 0.69), suggesting satisfactory effectiveness of our blinding protocol. Similarly, active guesses between the two groups were not different at the end of treatment (p = 0.53)
Comment:4.
Was the presence of negative symptoms and their severity part of the inclusion criteria?
Response:4.
Thank for the valuable opinion from the reviewer. We did not have the presence of negative symptoms and their severity as part of the inclusion criteria. We discussed this in the limitations on page 15 of the manuscript.
First, our trial did not explicitly target negative symptoms due to the inclusion of a small sample of patients not fulfilling the criteria of predominant negative symptoms
Comment:5.
Given the inclusion of patients with schizoaffective disorder, it would commenting on whether patients were excluded who were likely to have significant depressive symptoms (these are not really well assessed on the PANSS emotional/depressed subscale) prominent mood symptoms
Response:5.
Thank for the valuable opinion from the reviewer. As suggested, we a comment on this on page 3 of the manuscript.
The exclusion criteria were: (1) Having unstable medical conditions, current psychiatric comorbidity, prominent mood symptoms or active substance use disorder (in exception to caffeine and/or tobacco);
Comment:6.
How much payment was provided?
Response:6.
As suggested, we add the amount of payment on page 6 of the manuscript.
The payment (17.95 $) was offered to research participants to reimburse them for the transportation expense of each visit.
Comment:7.
The use if the HRV biomarker requires some justification
Response:7
Thank for the valuable opinion from the reviewer. As suggested, we add the justification for HRV measurement on page 6 of the manuscript.
Studies have reported that HRV is significantly correlated both with negative symptoms [21] and with frontoparietal network connectivity [22, 23].
Comment:8.
A mixed model approach would have been better repeated measures one but given the dropout rates I think this is oka
Response:8
Thank for the valuable opinion from the reviewer.
Reviewer 2 Report
In this paper, the authors showed that online frontoparietal theta-tACS during WM training can improve negative symptoms and other clinical outcomes, possibly through modulating the left frontoparietal network. This work is important as it can open a new area of tACS research for negative symptoms in schizophrenia. This paper is written in professional English, with sufficient introduction, detailed methods and solid data. However, the discussion is relatively superficial and can be further improved. The authors should discuss more on the neural circuits where frontoparietal network is involved in the symptoms in schizophrenia. Previous studies have demonstrated that the prefrontal cortex and posterior parietal cortex are mutually and extensively connected with the thalamus, the pulvinar nucleus in particular (Roth et al., 2016). Pulvinar plays a very important role in the contextual and multi-sensory processing (Chou et al., 2020; Fang et al., 2020; Ibrahim et al., 2016) and also emotional response, given its projection to the amygdala (Zhou et al., 2018). Given its strategic position in the sensory and emotional processing, pulvinar has been proved to be involved in the symptoms of schizophrenia (Shen et al., 2021; Cho et al., 2019). The authors should include these key citations in the discussion section and elaborate more on the function of pulvinar as a subcortical hub for the functioning of the frontoparietal network in normal brain and schizophrenia. I would like to recommend the revised paper to the editors for publication.
Roth, M. M., Dahmen, J. C., Muir, D. R., Imhof, F., Martini, F. J., & Hofer, S. B. (2016). Thalamic nuclei convey diverse contextual information to layer 1 of visual cortex. Nature neuroscience, 19(2), 299–307. https://doi.org/10.1038/nn.4197
Chou, X. L., Fang, Q., Yan, L., Zhong, W., Peng, B., Li, H., Wei, J., Tao, H. W., & Zhang, L. I. (2020). Contextual and cross-modality modulation of auditory cortical processing through pulvinar mediated suppression. eLife, 9, e54157. https://doi.org/10.7554/eLife.54157
Fang, Q., Chou, X. L., Peng, B., Zhong, W., Zhang, L. I., & Tao, H. W. (2020). A Differential Circuit via Retino-Colliculo-Pulvinar Pathway Enhances Feature Selectivity in Visual Cortex through Surround Suppression. Neuron, 105(2), 355–369.e6. https://doi.org/10.1016/j.neuron.2019.10.027
Ibrahim, L. A., Mesik, L., Ji, X. Y., Fang, Q., Li, H. F., Li, Y. T., Zingg, B., Zhang, L. I., & Tao, H. W. (2016). Cross-Modality Sharpening of Visual Cortical Processing through Layer-1-Mediated Inhibition and Disinhibition. Neuron, 89(5), 1031–1045. https://doi.org/10.1016/j.neuron.2016.01.027
Zhou, N., Masterson, S. P., Damron, J. K., Guido, W., & Bickford, M. E. (2018). The Mouse Pulvinar Nucleus Links the Lateral Extrastriate Cortex, Striatum, and Amygdala. The Journal of neuroscience : the official journal of the Society for Neuroscience, 38(2), 347–362. https://doi.org/10.1523/JNEUROSCI.1279-17.2017
Shen, L., Liu, D., & Huang, Y. (2021). Hypothesis of subcortical visual pathway impairment in schizophrenia. Medical hypotheses, 156, 110686. https://doi.org/10.1016/j.mehy.2021.110686
Cho, K., Kwak, Y. B., Hwang, W. J., Lee, J., Kim, M., Lee, T. Y., & Kwon, J. S. (2019). Microstructural Changes in Higher-Order Nuclei of the Thalamus in Patients With First-Episode Psychosis. Biological psychiatry, 85(1), 70–78. https://doi.org/10.1016/j.biopsych.2018.05.019
Author Response
Reviewer 2
Comments and Suggestions for Authors
In this paper, the authors showed that online frontoparietal theta-tACS during WM training can improve negative symptoms and other clinical outcomes, possibly through modulating the left frontoparietal network. This work is important as it can open a new area of tACS research for negative symptoms in schizophrenia. This paper is written in professional English, with sufficient introduction, detailed methods and solid data. However, the discussion is relatively superficial and can be further improved. The authors should discuss more on the neural circuits where frontoparietal network is involved in the symptoms in schizophrenia. Previous studies have demonstrated that the prefrontal cortex and posterior parietal cortex are mutually and extensively connected with the thalamus, the pulvinar nucleus in particular (Roth et al., 2016). Pulvinar plays a very important role in the contextual and multi-sensory processing (Chou et al., 2020; Fang et al., 2020; Ibrahim et al., 2016) and also emotional response, given its projection to the amygdala (Zhou et al., 2018). Given its strategic position in the sensory and emotional processing, pulvinar has been proved to be involved in the symptoms of schizophrenia (Shen et al., 2021; Cho et al., 2019). The authors should include these key citations in the discussion section and elaborate more on the function of pulvinar as a subcortical hub for the functioning of the frontoparietal network in normal brain and schizophrenia. I would like to recommend the revised paper to the editors for publication.
Roth, M. M., Dahmen, J. C., Muir, D. R., Imhof, F., Martini, F. J., & Hofer, S. B. (2016). Thalamic nuclei convey diverse contextual information to layer 1 of visual cortex. Nature neuroscience, 19(2), 299–307. https://doi.org/10.1038/nn.4197
Chou, X. L., Fang, Q., Yan, L., Zhong, W., Peng, B., Li, H., Wei, J., Tao, H. W., & Zhang, L. I. (2020). Contextual and cross-modality modulation of auditory cortical processing through pulvinar mediated suppression. eLife, 9, e54157. https://doi.org/10.7554/eLife.54157
Fang, Q., Chou, X. L., Peng, B., Zhong, W., Zhang, L. I., & Tao, H. W. (2020). A Differential Circuit via Retino-Colliculo-Pulvinar Pathway Enhances Feature Selectivity in Visual Cortex through Surround Suppression. Neuron, 105(2), 355–369.e6. https://doi.org/10.1016/j.neuron.2019.10.027
Ibrahim, L. A., Mesik, L., Ji, X. Y., Fang, Q., Li, H. F., Li, Y. T., Zingg, B., Zhang, L. I., & Tao, H. W. (2016). Cross-Modality Sharpening of Visual Cortical Processing through Layer-1-Mediated Inhibition and Disinhibition. Neuron, 89(5), 1031–1045. https://doi.org/10.1016/j.neuron.2016.01.027
Zhou, N., Masterson, S. P., Damron, J. K., Guido, W., & Bickford, M. E. (2018). The Mouse Pulvinar Nucleus Links the Lateral Extrastriate Cortex, Striatum, and Amygdala. The Journal of neuroscience : the official journal of the Society for Neuroscience, 38(2), 347–362. https://doi.org/10.1523/JNEUROSCI.1279-17.2017
Shen, L., Liu, D., & Huang, Y. (2021). Hypothesis of subcortical visual pathway impairment in schizophrenia. Medical hypotheses, 156, 110686. https://doi.org/10.1016/j.mehy.2021.110686
Cho, K., Kwak, Y. B., Hwang, W. J., Lee, J., Kim, M., Lee, T. Y., & Kwon, J. S. (2019). Microstructural Changes in Higher-Order Nuclei of the Thalamus in Patients With First-Episode Psychosis. Biological psychiatry, 85(1), 70–78. https://doi.org/10.1016/j.biopsych.2018.05.019
Response
Thank for the valuable opinion from the reviewer. As suggested, we revise the manuscript that can be seen on page 14.
Additionally, research has demonstrated that the prefrontal cortex and posterior parietal cortex are mutually and extensively connected with the thalamus, the pulvinar nucleus in particular [41]. In normal brains, pulvinar serves as a subcortical hub for the functioning of the frontoparietal network and plays an important role in the contextual and multi-sensory processing [42-44] and also emotional response, given its projection to the amygdala [45]. Given its strategic position in the sensory and emotional processing, pul-vinar has been proved to be involved in the psychopathological symptoms of schizo-phrenia, e.g., impairment of sensory processing and spatial working memory [46, 47]. It is possible that online theta-tACS stimulating the key nodes in the frontoparietal network may normalize the functional connectivity between the cortical and subcortical hub (i.e., pulvinar) regions and thereby contribute to its clinical efficacy.
Round 2
Reviewer 1 Report
All my concerns have been appropriately addressed.